DATA RELEASE

# The genome assembly and annotation of the white-lipped tree pit viper *Trimeresurus albolabris*

Xiaotong Niu[1,2,3,†], Yakui Lv[4,†], Jin Chen[5,†], Yueheng Feng[1], Yilin Cui[1], Haorong Lu[5,*] and Hui Liu[1,*]

1 Key Laboratory of Genetics and Germplasm Innovation of Tropical Special Forest Trees and Ornamental Plants (Ministry of Education), School of Tropical Agriculture and Forestry (School of Agricultural and Rural Affairs, School of Rural Revitalization), Hainan University, Haikou, 570228, China
2 State Key Laboratory of Agricultural Genomics, BGI-Shenzhen, Shenzhen, 518083, China
3 School of Ecology, Sun Yat-sen University, Shenzhen, 510275, China
4 College of Ecology and Environment, Hainan University, Haikou, 570228, China
5 China National GeneBank, BGI-Shenzhen, Shenzhen, 518120, China

## ABSTRACT

*Trimeresurus albolabris*, also known as the white-lipped pit viper or white-lipped tree viper, is a highly venomous snake distributed across Southeast Asia and the cause of many snakebite cases. In this study, we report the first whole genome assembly of *T. albolabris* obtained with next-generation sequencing from a specimen collected in Mengzi, Yunnan, China. After genome sequencing and assembly, the genome of this male *T. albolabris* individual was 1.51 Gb in length and included 38.42% repeat-element content. Using this genome, 21,695 genes were identified, and 99.17% of genes could be annotated using gene functional databases. Our genome assembly and annotation process was validated using a phylogenetic tree, which included six species and focused on single-copy genes of nuclear genomes. This research will contribute to future studies on *Trimeresurus* biology and the genetic basis of snake venom.

**Submitted:** 07 July 2023

\* Corresponding authors. E-mail: luhaorong@genomics.cn; liuhui@hainanu.edu.cn

† Contributed equally.

Preprint submitted at https://doi.org/10.20944/preprints202401.1712.v1

Included in the series: ***Snake Genomes*** (https://doi.org/10.46471/GIGABYTE_SERIES_0004)

**Subjects** Genetics and Genomics, Evolutionary Biology, Zoology

## INTRODUCTION

*Trimeresurus albolabris*, also known as the white-lipped pit viper, white-lipped tree viper, white-lipped bamboo pit viper, and green tree pit viper, is a venomous snake species belonging to the family Viperidae [1]. It is a relatively small snake, with adults typically measuring around 70–90 cm in length, and is known for its distinctive appearance, with a white stripe running down the center of its upper lip [2] (Figure 1). This species has been reported in China, Vietnam, Thailand, Laos, Cambodia, India, Bangladesh, Myanmar, and West Java and has become one of the most common venomous snakes with medical importance in Southeast Asia [3]. *T. albolabris* is a highly venomous snake. Its bite can be dangerous to humans, causing symptoms ranging from pain and swelling to more severe ones, such as shock, spontaneous bleeding, defibrination, and other complications of thrombocytopenia and leukocytosis [4, 5]. Notably, the venom of *T. albolabris* contains metalloproteinases [6, 7], a thrombin-like enzyme [8], and other venom components [5, 9].

## CONTEXT

Despite its venomous nature, *T. albolabris* is also an important research subject for its sexual dimorphism [10] and geographic variation [11]. A complete and high-quality genome

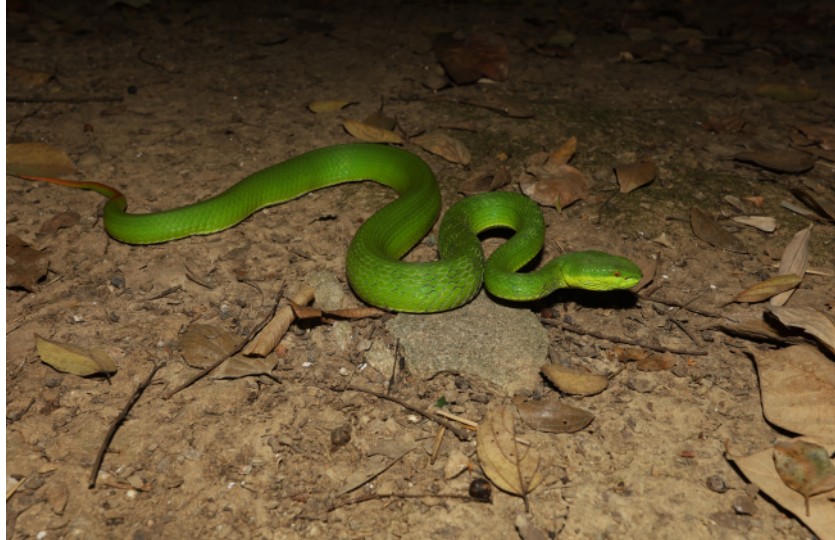

**Figure 1.** *Trimeresurus albolabris*, also known as the white-lipped pit viper, photographed by Diancheng Yang.

of this species is crucial for studying venom proteomics, particularly for drug discovery, developing antivenom therapies, and understanding the evolution of venomous species [12–14]. However, a complete genome of *T. albolabris* has not been published yet [15].

Here, we report the first whole genome with high continuity of a male *T. albolabris* individual, collected from Mengzi, Yunnan, China. The genome was generated using single-tube long fragment read (stLFR) [16] and whole genome sequencing (WGS) technologies. Our *T. albolabris* genome had a repeat element content of 38.42% and a total size of 1.51 Gb. This new genome assembly provides valuable evidence for future studies on snake venom and the genetic underpinnings of the *Trimeresurus* species.

## METHOD

The detailed stepwise protocols used in this study are gathered in a protocols.io collection, with the minor adaptations outlined below (Figure 2) [17].

### Sample collection and sequencing

A male *T. albolabris* sample was captured in Mengzi, Yunnan, China. To preserve its quality, this specimen was frozen in dry ice (at −80 °C) immediately after collection and identification, both for storage and transportation. The protocols we used for DNA extraction, library construction, and sequencing can be found in a protocols.io protocol collection [17]. The heart, stomach, liver, and kidneys were used for RNA sequencing. Additionally, a muscle sample was used for stLFR and WGS sequencing. The genome assembly and annotation workflow is also included in the same protocols.io protocol [17].

This study, including sample collection, experimental procedures, and research design, was approved by the Institutional Review Board of Beijing Genomics Institute (BGI-IRB E22017). Throughout the research, meticulous adherence to the guidelines established by BGI-IRB was strictly followed, ensuring compliance with ethical and regulatory standards.

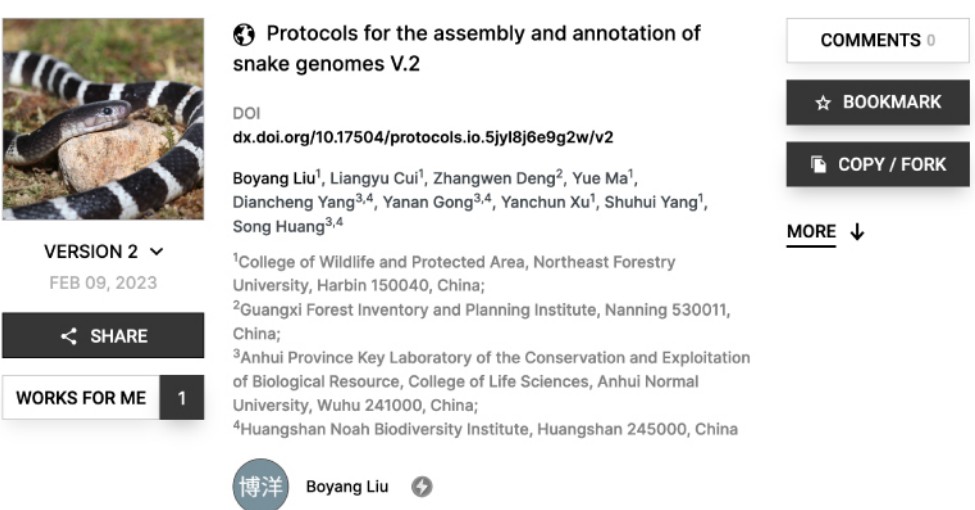

**Figure 2.** A protocols.io collection of protocols for sequencing snake genomes [17]. https://www.protocols.io/widgets/doi?uri=dx.doi.org/10.17504/protocols.io.4r3l27ez4g1y/v1

## Genome assembly, annotation, and assessment

The stLFR sequencing data were subjected to assembly using Supernova [18] (v2.1.1, RRID:SCR_016756). Subsequently, the gap-filling and redundancy removal steps were performed using GapCloser [19] (v1.12-r6, RRID:SCR_015026) and redundans [20] (v0.14a), respectively. These processes involved utilizing the WGS data to address the gaps in the assembly and eliminate redundant sequences.

To identify known repeat elements within the genome sequences, several genomics tools were employed, including Tandem Repeats Finder [21] (v. 4.09, RRID:SCR_022193), LTR_Finder [22] (RRID:SCR_015247), RepeatModeler [23] (v1.0.8), RepeatMasker [24] (v. 3.3.0, RRID:SCR_015027), and RepeatProteinMask (v. 3.3.0) [25]. For the prediction of protein-coding genes, we used a comprehensive approach combining *de novo*, homology-based, and transcript mapping strategies. *De novo* gene prediction was performed using GlimmerHMM [26] (RRID:SCR_002654). For RNA-seq-based predictions, RNA-seq data was first filtered by Trimmomatic [27] (v0.30, RRID:SCR_011848). After obtaining clean RNA-seq data, transcripts were assembled using Trinity [28] (v2.13.2, RRID:SCR_013048). Finally, PASA [29] (v2.0.2, RRID:SCR_014656) was used to align transcripts against the white-lipped tree viper genome to obtain gene structures. Homology-based prediction was performed by mapping protein sequences of the UniProt database (release-2020_05), *Pseudonaja textilis* (GCA_900518735.1), *Protobothrops mucrosquamatus* (GCA_001527695.3), *Thamnophis elegans* (GCA_009769535.1), and *Notechis scutatus* (GCA_900518725.1) to the white-lipped tree viper genome using Blastall (v2.2.26) [30] with an E-value cut-off of 1e-5. Next, we used GeneWise [31] (v2.4.1, RRID:SCR_015054) to analyze the alignment results and predict gene homology. Finally, the integration of RNA-seq, homology, and *de novo* predicted genes resulted in the generation of a final gene set using the MAKER pipeline (v3.01.03, RRID:SCR_005309) [32]. This approach, incorporating multiple genomic tools and techniques, facilitated the annotation and prediction of genes in the white-lipped tree viper genome.

**Table 1.** Summary statistics of *T. albolabris* of WGS paired-end (fq - fastq 1 and fastq 2) sequenced reads.

| | WGS-1 | | WGS-2 | | WGS-3 | |
|---|---|---|---|---|---|---|
| | **fq1** | **fq2** | **fq1** | **fq2** | **fq1** | **fq2** |
| %Q20 | 96.98 | 97.81 | 97.74 | 94.96 | 95.69 | 97.59 |
| %Q30 | 90.79 | 90.6 | 92.83 | 84.27 | 84.46 | 89.81 |
| %GC | 40.37 | 40.21 | 41.02 | 40.81 | 40.36 | 40.47 |
| %ErrorRate | 0.351809 | 0.233019 | 0.264481 | 0.540272 | 0.449364 | 0.257064 |
| TotalReads | 492,445,828 | | 425,689,572 | | 104,911,172 | |
| TotalBases | 98,489,165,600 | | 85,137,914,400 | | 20,982,234,400 | |

**Table 2.** Summary statistics of *T. albolabris* stLFR and RNA sequenced reads.

| | stLFR-1 | | stLFR-2 | | RNA-seq | |
|---|---|---|---|---|---|---|
| | **fq1** | **fq2** | **fq1** | **fq2** | **fq1** | **fq2** |
| %Q20 | 96.53 | 95.59 | 96.41 | 96.3 | 98.3 | 98.19 |
| %Q30 | 89.83 | 87.26 | 87.98 | 86.37 | 94.3 | 93.71 |
| %GC | 39.34 | 42.16 | 39.28 | 42.15 | 44.11 | 44.07 |
| %ErrorRate | 0.403065 | 0.525486 | 0.442228 | 0.392415 | 0.194523 | 0.205665 |
| TotalReads | 633,976,833 | | 161,105,172 | | 50,828,075 | |
| TotalBases | 145,814,671,590 | | 37,054,189,560 | | 10,165,615,000 | |

Functional annotation was performed using BLAST search, comparing with several databases, including SwissProt, TrEMBL, and Kyoto Encyclopedia of Genes and Genomes (KEGG), and limiting the E-value cut-off to 1e-5. InterProScan [26] (v5.52-86.0, RRID:SCR_005829) was used to predict motifs and domains, as well as gene ontology (GO) terms.

Benchmarking Universal Single-Copy Orthologs (or BUSCO, v5.2.2, RRID:SCR_015008) with genome mode and lineage data from vertebrata_odb10 was used to evaluate the completeness of our genome [33].

A reconstructed phylogenetic tree was generated by OrthoFinder (v2.3.7, RRID:SCR_017118) [34], which can search for single-copy orthologs among the protein sequences of *Chelonia mydas* (GCA_015237465.2), *Gallus gallus* (GCA_016699485.1), *Homo sapiens* (GCA_000001405.29), *Mus musculus* (GCA_000001635.9), *Ophiophagus hannah* (GCA_000516915.1), *Python bivittatus* (GCA_000186305.2), *Xenopus tropicalis* (GCA_000004195.4), *Alligator mississippiensis* (GCA_000281125.4), *Danio rerio* (GCA_000002035.4), *Anolis carolinensis* (GCA_000090745.2), *Gopherus evgoodei* (GCA_007399415.1), *Podarcis muralis* (GCA_004329235.1), and *Deinagkistrodon acutus* [35].

## RESULTS

This study on snake genomics resulted in a total of 387.48 Gb of paired-end (fastq 1 and fastq 2) data, which comprised 204.61 Gb of short reads data obtained through WGS sequencing and 182.87 Gb of long reads data obtained through stLFR sequencing, as shown in Tables 1 and 2.

We generated the first whole genome assembly of *T. albolabris* with high continuity, with a total genome size of 1.51 Gb, 39.97% GC content, and a scaffold N50 length of 381.55 kb (Table 3). The assembled *T. albolabris* genome consists of 10,016 contigs over 1,000 base pairs, with a total length of 1.50 Gb, accounting for 99.14% of the genome's total length. This resource will provide valuable evidence to explore new perspectives in the study of the *Trimeresurus* viper genomics.



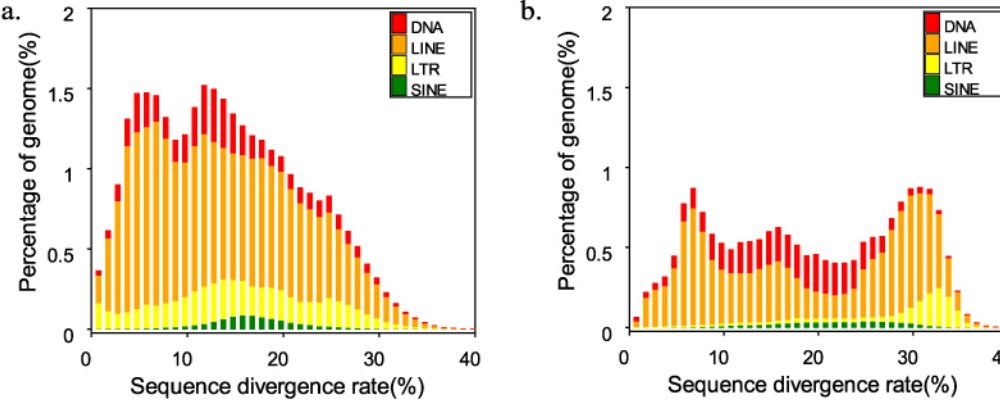

**Figure 3.** Distribution of transposable elements (TEs) in our *T. albolabris* genome. The TEs include DNA transposons (here indicated as DNA) and RNA transposons (i.e., DNAs, LINEs, LTRs, and SINEs). (a) Distribution of the *de novo* sequence divergence-rate. (b) Distribution of the known sequence divergence-rate.

**Table 3.** Summary of the features of the *T. albolabris* genome.

|  | Contigs | Contigs > (1,000 bp) | Contigs > (10,000 bp) |
|---|---|---|---|
| Total number ( >) | 71,131 | 46608 | 10,016 |
| Total length of (bp) | 1,513,852,334 | 1,501,212,553 | 1,355,102,082 |
| N50 Length (bp) |  | 381,553 |  |
| N75 Length (bp) |  | 115,212 |  |
| GC content is (%) |  | 39.97 |  |

**Table 4.** Statistics of the repetitive sequences identified in our *T. albolabris* genome.

| Type | Repeat size | % of genome |
|---|---|---|
| Trf | 47,767,541 | 3.155363 |
| Repeatmasker | 252,985,952 | 16.711402 |
| Proteinmask | 185,792,360 | 12.272819 |
| *De novo* | 498,353,737 | 32.919574 |
| Total | 581,568,803 | 38.416482 |

We detected repetitive elements in the *T. albolabris* genome, accounting for 38.42% of the total genome. Among them, the highest proportion was occupied by long interspersed nuclear elements (LINEs), which accounted for 23.94% and amounted to approximately 362.35 Mb. These findings were found to be highly similar to the repetitive element content observed in previously sequenced genomes, such as those of *Thamnophis elegans* (42.02%) (accession No. PRJNA561996) and *Crotalus tigris* (42.31%) [36]. This indicates that the results we obtained are highly reliable and plausible. The remaining types of transposable elements, including DNA transposons, long terminal repeats (LTRs), and short interspersed nuclear elements (SINEs), accounted for 6.90%, 5.83%, and 1.24%, respectively (Figure 3, Tables 4, and 5).

Using homology-based, *de novo*, and RNA-sequencing annotation methods, we successfully identified 21,695 protein-coding genes in our *T. albolabris* genome assembly. We compared our assembly to those of *Notechis scutatus* (GCA_900518725.1), *Pseudonaja textilis* (GCA_900518735.1), and *Thamnophis elegans* (GCA_009769535.1), all of which are available from the NCBI database. Our analysis revealed no significant differences in the



**Table 5.** Summary of the TEs in our *T. albolabris* genome.

| Type | Repbase TEs | | TE protiens | | *De novo* | | Combined TEs | |
|---|---|---|---|---|---|---|---|---|
| | Length (bp) | % in genome | Length (bp) | % in genome | Length (bp) | % in genome | Length (bp) | % in genome |
| DNA | 51,357,881 | 3.392529 | 2,032,636 | 0.134269 | 64,605,374 | 4.267614 | 104,513,127 | 6.903786 |
| LINE | 184,866,441 | 12.211656 | 157,414,659 | 10.398284 | 294,829,469 | 19.475444 | 362,351,919 | 23.935751 |
| SINE | 9,622,825 | 0.635651 | 0 | 0 | 13,144,889 | 0.868307 | 18,769,499 | 1.23985 |
| LTR | 23,685,560 | 1.564589 | 26,413,572 | 1.744792 | 74,868,256 | 4.945546 | 88,305,953 | 5.833195 |
| Other | 77,658 | 0.00513 | 141 | 0.000009 | 0 | 0 | 77,799 | 0.005139 |
| Unknown | 0 | 0 | 0 | 0 | 98,895,691 | 6.532717 | 98,895,691 | 6.532717 |
| Total | 252,985,952 | 16.711402 | 185,792,360 | 12.272819 | 496,342,637 | 32.786727 | 566,552,754 | 37.424572 |

**Table 6.** Statistics for the miRNA, tRNA, rRNA, and snRNA discerned from our *T. albolabris* genome.

| Type | Copy (w) | Average length (bp) | Total length (bp) | % of genome |
|---|---|---|---|---|
| miRNA | 250 | 98.992 | 24,748 | 0.001635 |
| tRNA | 179 | 75.70949721 | 13,552 | 0.000895 |
| rRNA | 104 | 137.6057692 | 14,311 | 0.000945 |
| snRNA | 301 | 115.1229236 | 34,652 | 0.002289 |

**Table 7.** Consequences of gene functional annotation.

| Values | Total | Swissprot-annotated | KEGG-annotated | TrEMBL-annotated | Interpro-annotated | GO-annotated | Overall |
|---|---|---|---|---|---|---|---|
| Number | 21,695 | 20,240 | 19,216 | 21,134 | 21,019 | 14,786 | 21,516 |
| Percentage | 100% | 93.29% | 88.57% | 97.41% | 96.88% | 68.15% | 99.17% |

distribution of transcript mapping lengths, coding sequences (CDS) lengths, or the quantity of exons and introns. Additionally, our analysis predicted the presence of 250 miRNAs, 179 tRNAs, and 301 snRNAs within the *T. albolabris* genome (Table 6).

Comparing our results with various public datasets, such as InterPro [37], KEGG [38], SwissProt [39], TrEMBL [39], and GO terms, we identified 21,695 expanded gene families, including 99.17% functionally annotated genes (Table 7).

Further analyses using KEGG enrichment revealed that Environmental Information Processing, Organismal Systems, and Metabolism pathways were the most abundant, with Signal Transduction pathways being the most prominent. Among the Organismal Systems pathways, 1,774 Immune System genes and 1,551 Endocrine System genes were the most abundant (Figure 4a). In addition, based on the results of our GO analysis, we found that 7,900 genes are related to binding, while 7,740 genes are related to cellular processes (Figure 4b).

## DATA VALIDATION AND QUALITY CONTROL

We employed BUSCO v5.2.2 to assess the quality and completeness of our genome assembly [40]. The results of our BUSCO analysis revealed that our assembly achieved 85.3% completeness when evaluated against the vertebrata_odb10 database (Figure 5), indicating that our assembly is of relatively high quality and completeness.

To assess the quality of our assembly, we constructed a phylogenetic tree using the protein sequences of seven different amphibian and reptile species (*Anolis carolinensis, Chelonia mydas, Deinagkistrodon acutus, Ophiophagus hannah, Python bivittatus, Xenopus tropicalis, and Alligator mississippiensis*) as well as the protein sequences of *Gallus gallus, Homo sapiens, Mus musculus,* and *Danio rerio* downloaded from NCBI. The resulting

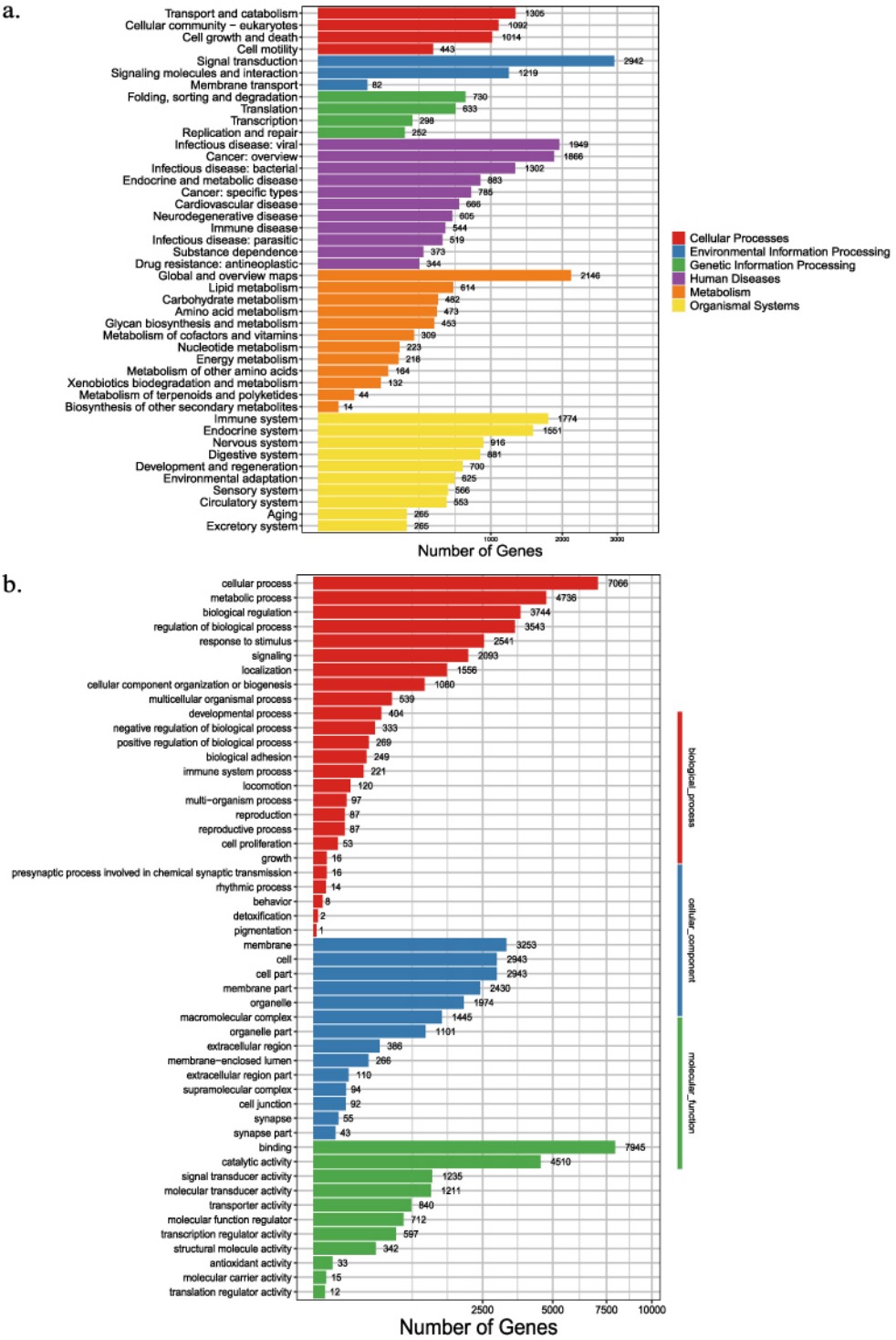

**Figure 4.** Gene annotation information obtained from our *T. albolabris* genome. (a) KEGG enrichment. (b) GO enrichment.

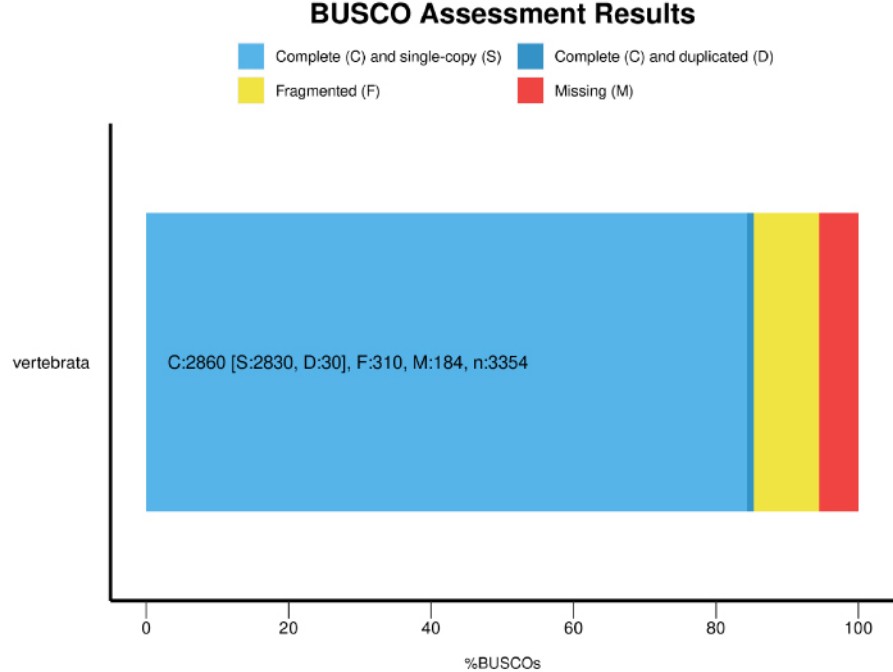

**Figure 5.** BUSCO Assessment result of our *T. albolabris* genome.

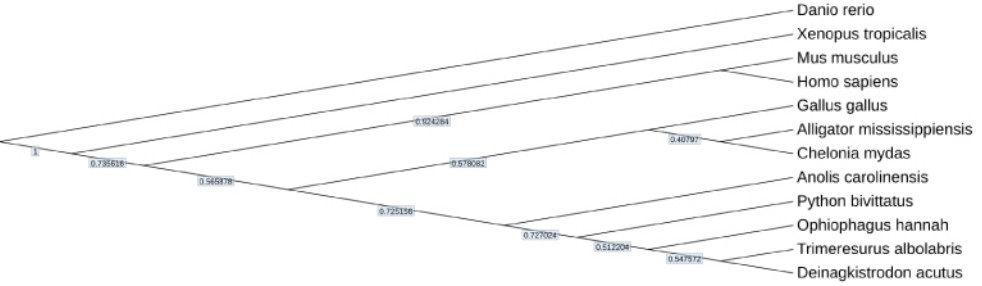

**Figure 6.** Phylogenetic tree reconstructed using nuclear genome single-copy genes. The numbers on the branches of the phylogenetic tree represent the branch lengths obtained using OrthoFinder.

phylogenetic tree is consistent with previous research, indicating that our data can accurately identify related species (Figure 6).

## REUSE POTENTIAL

We presented the first genome assembly of the white-lipped tree pit viper. This data provides new resources for studying the vipers biology and evolution, as well as the genetic foundation of its venom.

## DATA AVAILABILITY

The data supporting the findings of this study have been deposited into the CNGB Sequence Archive (CNSA) [41] of the China National GeneBank DataBase (CNGBdb) [42] with accession number CNP0004151. The raw data is also available in NCBI with the bioproject

number PRJNA955401 (see also the machine readable nanopublication: RAV3oIcruk).
Additional data is also available in GigaDB [43].

## EDITOR'S NOTE

This paper is part of a series of Data Release papers presenting the genomes of different
snake species [44].

## ABBREVIATIONS

fq: fastq; GO: gene ontology; KEGG: Kyoto Encyclopedia of Genes and Genomes; LINE: long
interspersed nuclear element; LTR: long terminal repeat; SINE: short interspersed nuclear
element; stLFR: single-tube long fragment read; TEs: transposable elements; WGS: whole
genome sequencing.

## DECLARATIONS

### Ethics approval

This study was approved by the Institutional Review Board of BGI (BGI-IRB E22017).

### Competing interests

The authors declare no competing interests.

### Author contribution

H Liu and H Lu designed and initiated the project. YC and YF performed the DNA extraction
and the library construction. XN and JC performed the data analysis. XN, JC and YL wrote
the manuscript. All authors read and approved the final manuscript.

### Acknowledgements

Our project was supported by the China National GeneBank (CNGB). This work was also
supported by Hainan University and BGI-Shenzhen. The Anhui Normal University collected
the samples. This work was supported by the Guangdong Provincial Key Laboratory of
Genome Read and Write (grant no. 2017B030301011).

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
