## [Editor Report]

Editor’s AssessmentThe white-lipped bamboo pit viper (Trimeresurus albolabris), is a venomous snake species distributed across SE and E Asia, and responsible for a lot of the snakebites in its range. To better understand the snake venom the genome of a male individual was sequenced, producing a 1.51 Gb sized assembly with 21,695 genes identified. During peer review more details on the methods were added and the figures improved. This data provides new resources for studying the viper’s biology and evolution, as well as the genetic foundation of its venom.

---

## [Reviewer Report]

Upload additional filesDRR-202307-01/form/Comments for gx-DR-1690463537.docxReviewer name and names of any other individual's who aided in reviewer Jing LiuDo you understand and agree to our policy of having open and named reviews, and having your review included with the published papers. (If no, please inform the editor that you cannot review this manuscript.)YesIs the language of sufficient quality?YesPlease add additional comments on language quality to clarify if needed
Are all data available and do they match the descriptions in the paper? YesAdditional CommentsAre the data and metadata consistent with relevant minimum information or reporting standards? See GigaDB checklists for examples <a href="http://gigadb.org/site/guide" target="_blank">http://gigadb.org/site/guide</a>YesAdditional CommentsIs the data acquisition clear, complete and methodologically sound?YesAdditional CommentsIs there sufficient detail in the methods and data-processing steps to allow reproduction?YesAdditional CommentsIs there sufficient data validation and statistical analyses of data quality? YesAdditional CommentsIs the validation suitable for this type of data?YesAdditional CommentsIs there sufficient information for others to reuse this dataset or integrate it with other data?YesAdditional CommentsAny Additional Overall Comments to the AuthorRecommendationAccept

---

## [Reviewer Report]

Upload additional filesDRR-202307-01/form/GIGABYTE-DRR-202307-01 - Review (1).pdfReviewer name and names of any other individual's who aided in reviewer M. EscalonaDo you understand and agree to our policy of having open and named reviews, and having your review included with the published papers. (If no, please inform the editor that you cannot review this manuscript.)YesIs the language of sufficient quality?YesPlease add additional comments on language quality to clarify if needed
Are all data available and do they match the descriptions in the paper? YesAdditional CommentsSee attachmentAre the data and metadata consistent with relevant minimum information or reporting standards? See GigaDB checklists for examples <a href="http://gigadb.org/site/guide" target="_blank">http://gigadb.org/site/guide</a>YesAdditional CommentsSee attachmentIs the data acquisition clear, complete and methodologically sound?NoAdditional CommentsSee attachmentIs there sufficient detail in the methods and data-processing steps to allow reproduction?NoAdditional CommentsSee attachmentIs there sufficient data validation and statistical analyses of data quality? YesAdditional CommentsSee attachmentIs the validation suitable for this type of data?YesAdditional CommentsSee attachmentIs there sufficient information for others to reuse this dataset or integrate it with other data?NoAdditional CommentsSee attachmentAny Additional Overall Comments to the AuthorSee attachmentRecommendationMinor Revision

---

## [Reviewer Report]

Reviewer name and names of any other individual's who aided in reviewer Chiaki KambayashiDo you understand and agree to our policy of having open and named reviews, and having your review included with the published papers. (If no, please inform the editor that you cannot review this manuscript.)YesIs the language of sufficient quality?YesPlease add additional comments on language quality to clarify if needed
Are all data available and do they match the descriptions in the paper? YesAdditional CommentsAre the data and metadata consistent with relevant minimum information or reporting standards? See GigaDB checklists for examples <a href="http://gigadb.org/site/guide" target="_blank">http://gigadb.org/site/guide</a>YesAdditional CommentsIs the data acquisition clear, complete and methodologically sound?YesAdditional CommentsIs there sufficient detail in the methods and data-processing steps to allow reproduction?YesAdditional CommentsIs there sufficient data validation and statistical analyses of data quality? YesAdditional CommentsIs the validation suitable for this type of data?YesAdditional CommentsIs there sufficient information for others to reuse this dataset or integrate it with other data?YesAdditional CommentsAny Additional Overall Comments to the AuthorThis manuscript reported data on the draft genome sequence of Trimeresurus albolabris, which will help to understand snake evolution and the genetic basis of snake venom.  Minor corrections are noted below: 1. Unification of unit notation ex) 1.51 GB (Page 1, Line 20) and 1.51G (Page 2, Line 21).  2. “Repeat Finder” seems to be an error for “Tandem Repeat Finder.” (Page 3, Line 23)  3. The average gene length seems to be missing (Page 6, Line 17).  4. Table 5 appears to be an error for Table 6 (Page 7, Line 6–7).  5. In Figure 6, the species listed in the manuscript (Page 9, Line 10–13) and the OTUs on the phylogenetic tree seem to be different. Also, it would be better to indicate what the numbers on the branches represent.
RecommendationMinor Revision